# Associations between Social Isolation Index and changes in grip strength, gait speed, bone mineral density (BMD), and self-reported incident fractures among older adults: Results from the Canadian Longitudinal Study on Aging (CLSA)

**Ahreum Lee**[1,2], **Caitlin McArthur**[3], **George Ioannidis**[1,2], **Alexandra Mayhew**[2,4], **Jonathan D. Adachi**[2], **Lauren E. Griffith**[2,4], **Lehana Thabane**[2,5], **Alexandra Papaioannou**[1,2] *

**1** Geras Centre for Aging Research, Hamilton, Ontario, Canada, **2** Department of Health Research Methods, Evidence and Impact, Master University, Hamilton, Ontario, Canada, **3** School of Physiotherapy, Dalhousie University, Halifax, Nova Scotia, Canada, **4** McMaster Institute for Research on Aging, Hamilton, Ontario, Canada, **5** Biostatistics Unit, St Joseph's Healthcare, Hamilton, Ontario, Canada

* papaioannou@hhsc.ca

## Abstract

### Background

The aim is to investigate whether social isolation and loneliness are associated with changes in grip strength, gait speed, BMD, and fractures.

### Methods

Canadian Longitudinal Study on Aging (CLSA) Comprehensive Cohort participants aged 65 years and older at baseline (2012–2015) who completed the three-year follow-up interview (2015–2018) were included in this analysis (n = 11,344). Social isolation and loneliness were measured using the CLSA social isolation index (CLSA-SII, range 0–10). We calculated absolute and percent change in grip strength (kg) and gait speed (m/s) and annualized absolute (g/cm$^2$) and percent change in femoral neck and total hip BMD during the three-year follow-up. Self-reported incident fractures of all skeletal sites in the previous 12 months were measured at three-year follow-up. Multivariable analyses were conducted. Odd ratio (OR) and 95% confidence interval (CI) are reported.

### Results

The mean age (standard deviation [SD]) was 72.9 (5.6) years and 49.9% were female. The mean (SD) of CLSA-SII at baseline was 3.5 (1.4). Mean absolute and percentage change (SD) in grip strength (kg) and gait speed (m/s) were -1.33 (4.60), -3.02% (16.65), and -0.05 (0.17), -3.06% (19.28) during the three-year follow-up, respectively. Mean annualized absolute (g/cm$^2$) and percentage change (SD) in femoral neck and total hip BMD were -0.004

**Data Availability Statement:** All relevant data are within the paper and its Supporting Information files.

**Funding:** The author(s) received no specific funding for this work.

**Competing interests:** The authors have declared that no competing interests exist.

(0.010), -0.47% (1.43) and -0.005 (0.009), -0.57% (1.09), respectively. 345 (3.1%) participants had incident fractures. As CLSA-SII increased (per one unit change), participants had 1.13 (adjusted OR 1.13, 95% CI 1.01–1.27) times greater odds for incident fractures. The interaction term between the CLSA-SII and centre for epidemiology studies depression 9 scale (CES-D 9) for self-reported incident fractures was shown (interaction OR 1.02, 95% CI 1.00–1.04).

## Conclusions

Socially isolated and lonely older adults were more likely to have had incident fractures, but social isolation was not associated with the three-year changes in grip strength, gait speed, or BMD.

## Introduction

Social isolation is an objective state delineated by lack of relationships, reduced network size, and limited social contacts [1, 2]. Loneliness is the subjective and psychological embodiment of social isolation with discrepancies between the frequency and intimacy of social contacts, and their current and desired relationships [2]. As age increases, older adults are more likely to experience social isolation or loneliness due to reduced relationships caused by retirement, physical disability, loss of friends/spouse or living alone [3], and decreased economic resources [1]. Twelve to nineteen percent of Canadians over the age of 65 years [4] and 24% of American community-dwelling adults aged 65 and older [5] experienced social isolation. Similarly, 43% of Americans aged 60 and older [5] and 20–34% of older adults in 25 European countries [6] reported being lonely.

Poor physical function, measured via grip strength and gait speed, are prevalent medical conditions which increase disability, frailty, dependence, and poor quality of life in older adults worldwide [7]. They may also lead to the development of serious musculoskeletal conditions such as sarcopenia and osteoporosis. Sarcopenia, defined as both weakness through low grip strength and slowness through low usual gait speed [8], is related to health issues such as falls, fractures, physical disability, poor quality of life, and mortality [9, 10]. Depending on the definition of sarcopenia, the prevalence ranges from 9.9% to 40.4% [11]. Osteoporosis is a condition in which the quantity and strength of bone decrease due to microarchitectural tissue deterioration, increasing the risk for fractures and mortality and decreasing quality of life [12, 13]. Osteoporosis is typically measured using bone mineral density (BMD).The prevalence of osteoporosis in Americans aged 50 years or older was 12.6% in 2017–2018 [14] and in Canadians aged 40 years or older was 11.9% in 2015–2016 [12].

Previous studies have examined the association between social isolation or loneliness and frailty, physical function, grip strength, BMD and falls, but the results are mixed. For example, social isolation was associated with poor physical function for community-dwelling older adults aged 70 years or older in a study from the United Kingdom [15] with frailty in a meta-analysis of three cross-sectional studies [16] and a subgroup analysis of males aged 60 years or older in the English Longitudinal Study on Aging (ELSA) [17]. In contrast, social isolation was not associated with areal BMD (g/cm$^2$) of the non-dominant hip in a study from the United Kingdom [15], frailty in a systematic review of longitudinal studies [16], or with 6-year change in frailty index in the ELSA study [17]. Likewise, though loneliness was associated with risk of

falls in a 5-year longitudinal cohort of American older adults [18], it was not associated with low grip strength in a 2-year longitudinal population-based cohort study of older adults from 10 European countries [19].

Despite the varied work to date, associations between social isolation and loneliness and longitudinal changes in grip strength, gait speed, BMD and fractures have not been explored. Further research addressing potential effects of social isolation and loneliness on musculoskeletal health for older adults is still needed due to the mixed results, lack of longitudinal studies, and the emerging importance of isolation during the COVID-19 pandemic. Therefore, our aim was to explore whether baseline social isolation index was associated with changes in grip strength, gait speed, BMD, and self-reported incident fractures among community-living older adults in Canada at the three-year follow-up.

## Methods

### The Canadian Longitudinal Study on Aging (CLSA)

The CLSA is a population-based study aimed at understanding the health status and behavior of older adults [20]. The CLSA includes a stratified random sample of community-dwelling males and females aged 45–85 years across 10 Canadian provinces [21] and consists of two cohorts: a comprehensive cohort (n = 30,097) and tracking cohort (n = 21,141). All CLSA participants provide extensive questionnaire data, and participants in the comprehensive cohort also undergo an in-depth physical assessment at one of the 11 data collection sites across Canada [21]. Biological, medical, psychological, social, lifestyle, and economic data are collected from participants every three years for 20 years starting in 2010–2015 [20, 21]. Participant exclusion criteria at recruitment included: 1) residing in the three Canadian territories, 2) living on federal First Nations reserves, 3) working full-time in the Canadians Armed Forces, 4) not being able to provide data in English or French, 5) not being able to provide data without a proxy (e.g., a cognitive impairment), and 6) living in institutions [20, 21].

**Study sample.** There were 30,097 participants in the 2015 CLSA comprehensive cohort, of which 12,646 participants aged 65 years or older, and of those, 11,344 participants (89.7%) had follow-up data at three-years and were included in this study.

### Social isolation and loneliness

Social isolation and loneliness were measured with the validated CLSA Social Isolation Index (CLSA-SII) [22, 23] at baseline. The CLSA-SII encompasses a total of 29 items in three domains of a social isolation and loneliness conceptual framework: 1) structural social isolation with objective measurement, which includes social network connection (n = 23): community/social participation (n = 8), social network size (n = 8), contact with network members (n = 5), living arrangement (n = 1) and marital status (n = 1) [22]; 2) functional social isolation with objective measurement, including qualitative and behavioral valuable components of social resources given or received by interactions between people (n = 4): the Medical Outcome Study Social Support Scale including affectionate, emotional/informational, positive social interaction and tangible support [22]; and 3) functional social isolation with subjective measurement [24], which is opportunity, sufficiency, and feelings related to the social connections (n = 2): loneliness and desire to participate in more activities. Our study followed the same procedure of constructing the CLSA-SII used in prior studies, whereby all included measures in the CLSA-SII were converted into scores between 0 and 10 for each of the three domains which were summed and divided by 3 to construct a final score that ranged from 0 to 10 [22, 23]. The CLSA-SII is an overall score representing social isolation and loneliness, not a

sub-score representing the factors included. Higher score in the CLSA-SII indicates greater social isolation and loneliness [22].

## Change in grip strength and gait speed at three-year follow-up

Grip strength (kg) was measured for each participant using their dominant hand over three attempts and the maximum value was used for analysis [25]. The 4-m gait speed (m/s) was measured by asking participants to walk 4-m at a regular pace, and was calculated by dividing 4-m by the walking time in seconds (m/s) [25]. Grip strength and gait speed were measured twice for each individual. In addition, grip strength and gait speed were measured in the same manner at baseline and three-year follow-up. Absolute change in grip strength (kg) and gait speed (m/s) was calculated by subtracting the baseline from the follow-up values. Percentage change (%) in grip strength and gait speed was calculated by dividing the absolute change by the value at baseline and multiplying by 100 (%).

## Change in BMD and osteoporosis classification by dual X-ray absorptiometry (DXA) at three-year follow-up

DXA images were measured for bone mineral content through Hologic Discovery A™. DXA scans were acquired in the same manner at baseline and three-year follow-up, and were centrally reanalyzed [26]. As the CLSA did not acquire lumbar spine and wrist BMD, only femoral neck and total hip BMD were used. For bone density, the smallest detectable difference in absolute measurements (g/cm$^2$) remains consistent regardless of the baseline BMD, while the percentage change is influenced by both the baseline bone density and age [27]. We assessed BMD changes by considering both absolute and percentage changes in BMD according to the International Society for Clinical Densitometry (ISCD) [28]. The absolute change represents the actual numeric variation in BMD over a specific period, while the percentage change is expressed as a ratio relative to the baseline BMD change rate, enabling the comparison of BMD changes among individuals with different baseline BMD values [28]. It is important to note that individuals with a very low baseline BMD may have a larger percentage change in BMD compared to those with a higher baseline BMD due to the same absolute BMD change [29]. Change in BMD was assessed by annualized absolute (g/cm$^2$) and percentage change in femoral neck and total hip. Annualized absolute (g/cm$^2$) change in femoral neck and total hip BMD was calculated as (follow-up BMD–baseline BMD)/3 (years). Annualized percentage change in femoral neck and total hip BMD was calculated as [(follow-up BMD–baseline BMD)/ baseline BMD] *100 (%)/3 (years). Osteoporosis classification by DXA was assessed according to the World Health Organization (WHO) definition [30]: osteoporosis (T-score≤-2.5), osteopenia (-1.0<T-score<-2.5), and normal bone density (T-score≥-1.0). Changes for osteoporosis classification from baseline to follow-up were categorized as follows: 1) worsened (from normal bone density to osteopenia, normal bone density to osteoporosis, or osteopenia to osteoporosis), 2) unchanged (from normal bone density to normal bone density, osteopenia to osteopenia, or osteoporosis to osteoporosis), and 3) improved (from osteopenia to normal bone density, osteoporosis to normal bone density, or osteoporosis to osteopenia).

## Change in self-reported incident fractures at three-year follow-up

All self-reported fracture data were collected over a one-year period. The time period for prior fractures was between baseline (2015) to one year prior (2015 to 2014). The time period for new fractures at the three-year follow-up was between year three (2018) and one year prior (2018 to 2017). Potential fractures that may have occurred between baseline and year 2 were not assessed (2015–2017). For both time points (baseline and three-year follow-up), self-

reported fractures were assessed by a question from the CLSA questionnaire that stated: "In the last 12 months, have you had any injuries by one or more broken or fractured bones that were serious enough to limit some of your normal activities?". Those who answered "yes" were assessed as having "self-reported fractures". All types of fractures were included regardless of skeletal sites, given that almost all participants did not identify the location of the skeletal site during the three-year follow-up assessment.

Self-reported incident fractures from baseline to post three-year were categorized as follows: 1) yes (from no prior fractures to incident fractures, or prior fractures to incident fractures) and 2) no (from prior fractures to no incident fractures, or no prior fractures to no incident fractures).

## Covariates

Based on the findings of previous epidemiological studies, we included the following baseline covariates related to the associations between baseline social isolation and loneliness and changes in grip strength, gait speed, BMD, and self-reported incident fractures at year 3: age group (65–74 years, ≥75 years), sex (males, females), education (less than secondary, secondary, post-secondary), body mass index (BMI, kg/m$^2$), total household income (less than $20,000, $20,000-$49,999, $50,000-$99,999, $100,000-$149,999, ≥$150,000), smoking status (current smoker, non-smoker, former smoker), alcohol consumption (almost every day, 2–5 times a week, 1–4 times a month, less than once a month, never), self-reported osteoporosis (yes, no), self-reported rheumatoid arthritis (yes, no), self-reported history of fractures since adulthood (yes, no), maternal fractures history (yes, no), corticosteroid use (yes, no), self-reported prior falls (yes, no), diabetes (yes, no), DXA femoral neck BMD T-score (SD), grip strength (kg), gait speed (m/s), the five-item satisfaction with life scale [(SWLS), range 5–35, where higher scores indicate greater life satisfaction] [31, 32], centre for epidemiology studies depression 9 Scale [(CES-D 9), range 0–27 where higher scores indicate greater depression] [33], psychological distress (range 10–43, where higher scores indicate higher mental distress) [34, 35], nutritional risk [(AB SCREEN II), range 0–48 where higher scores indicate a higher level of nutritional risk) [36], perceived mental health, perceived health, and physical activity scale for the elderly score (PASE) (range 0–629, where higher scores indicate greater physical activity [37]. Given that a single-item loneliness question that is a part of the CES-D 10 was included in the CLSA-SII instrument, a total of nine items (e.g., easily bothered and feel fearful or tearful) except for loneliness was included in CES-D 9 [22]. This was based on the evidence that SWLS, perceived health, perceived mental health and CES-D 9 correlated with the CLSA-SII from the previous study by Wister et al. [22]. In addition, we included other variables related to osteoporosis or fractures, such as self-reported osteoporosis and self-reported history of fractures since adulthood, as covariates.

## Statistical analysis

We expressed continuous and categorical data as mean [standard deviation (SD)] and frequency (percentage), respectively. We used the analytic weights based on CLSA's guidance documentation for analyzing complex survey data [38] for the multivariable linear/multivariable logistic/multinomial regression models.

We hypothesized that socially isolated and lonely older adults may have greater reductions in grip strength, gait speed, or (femoral neck and total hip) BMD or had an increased risk of fractures. We used T-Test (binary, dichotomy), Welch's ANOVA (category), and Pearson correlation (continuous) for univariate analyses of the CLSA-SII in all participants. To test these hypotheses, we conducted each of the following multivariable analyses: 1) independent

(exposure) variable: CLSA-SII (continuous), 2) outcome (response) variables: a) changes in grip strength (continuous, linear), b) changes in gait speed (continuous, linear), c) changes in BMD (continuous, linear) and change for osteoporosis classification by DXA (category, multinomial), d) self-reported incident fractures (binary, logistic). Model 1 was unadjusted and Model 2 was a fully adjusted multivariable linear or logistic and multinomial logistic regression model. In Model 2, all outcome variables were adjusted for all the same covariates. Model 3 was a fully adjusted multivariable linear and logistic regression model adding significant interaction terms to model 2. We reported coefficient (β) and 95% confidence interval (CI) through multiple linear regression models and odds ratio (OR) and 95% CI through multivariable logistic and multinomial logistic regression models.

We also conducted a subgroup analysis by sex for each association. Based on previous studies that showed social isolation/loneliness were associated with age, sex, depression, satisfaction with life [22], and physical activity [39], we tested two-way interactions between CLSA-SII and the following: age, sex, CES-D 9, SWLS, and PASE. We also tested collinearity between covariates (e.g., grip strength and gait speed).

We conducted multiple imputation for missing data (e.g., 13.6% of grip strength, gait speed, 10.9% of CLSA-SII, 10.3% of DXA femoral neck BMD and 8.6% of household income) and sensitivity analysis of the weighted sample without multiple imputation (i.e., complete cases).

All analyses were conducted under a significant level of 0.05 in two-sided tests using SAS v9.4 (The SAS Institute, Cary, NC).

### Ethics

This study utilized data collected by the CLSA, with the CLSA serving as the authorized data custodian and obtained written consent from all study participants. This study, which involved a secondary analysis of CLSA data, received approval from the Hamilton Integrated Research Ethics Board (Ethics Certificate No. 8253) under the application for "Retrospective Review of Medical Charts/Health Records" on December 6th, 2019. We used a de-identification record in order to prevent identification of individuals which was accessed on March 25th, 2020.

## Results

The mean age (SD) of all participants (n = 11,344) was 72.9 years (0.1) and 49.9% (n = 5,656) were females. The mean (SD) of CLSA-SII at baseline was 3.47 (1.40) (Table 1). Univariate analyses of the CLSA-SII in all participants are shown in S1 Table.

Mean (SD) absolute and percentage change from baseline in grip strength (kg) and gait speed (m/s) were -1.33kg (4.60), -3.02% (16.65), and -0.05m/s (0.17), -3.06% (19.28) at the three-year follow-up, respectively. Mean (SD) of annualized absolute (g/cm$^2$) and percentage change from baseline in femoral neck and total hip BMD were -0.004g/cm$^2$ (0.010) and -0.47% (1.43), and -0.005g/cm$^2$ (0.009) and -0.57% (1.09) at the three-year follow-up, respectively. At the three-year follow-up, 3.1% (n = 345) of participants experienced incident fractures while 8.6% (n = 765) of participants were exacerbated in the change for osteoporosis classification by DXA (Table 2).

Socially isolated and lonely older participants had a greater likelihood of self-reported incident fractures (unadjusted OR 1.20, 95% CI 1.08–1.34) (Model 1). In addition, socially isolated and lonely older participants were 1.13 times more likely to have self-reported incident fractures (adjusted OR 1.13, 95% CI 1.01–1.27) when adjusting for all covariates (Model 2). The associations between CLSA-SII and absolute and percentage change in grip strength and gait speed, annualized absolute and percentage changes in femoral neck and total hip BMD and

**Table 1. Characteristics of participants in CLSA at baseline and three-year follow-up.**

| Baseline characteristic | All (n = 11,344) | Males (n = 5,688) | Females (n = 5,656) |
|---|---|---|---|
| Age (year), mean (SD) | 72.9 (5.6) | 72.9 (5.6) | 72.9 (5.7) |
| Age group (year), n (%) | | | |
| 65–74 | 6,781 (59.8) | 3,404 (59.8) | 3,377 (59.7) |
| 75+ | 4,563 (40.2) | 2,284 (40.2) | 2,279 (40.3) |
| BMI (kg/m$^2$), mean (SD) | 27.8 (4.9) | 27.9 (4.2) | 27.7 (5.5) |
| Education, n (%) | | | |
| Less than secondary | 961 (8.5) | 393 (6.9) | 569 (10.1) |
| Secondary, no post-secondary | 1,219 (10.8) | 515 (9.1) | 704 (12.5) |
| Some post-secondary | 901 (8.0) | 417 (7.4) | 484 (8.5) |
| Post-secondary | 8,228 (72.8) | 4,338 (76.6) | 3,890 (68.9) |
| Total household income, n (%) | | | |
| Less than $20,000 | 672 (6.5) | 186 (3.5) | 484 (9.7) |
| $20,000-$49,999 | 3,393 (32.7) | 1,337 (25.0) | 2,056 (40.9) |
| $50,000-$99,999 | 4,200 (40.5) | 2,396 (44.8) | 1,804 (35.9) |
| $100,000-$149,999 | 1,390 (13.4) | 910 (17.0) | 480 (9.6) |
| $150,000+ | 713 (6.9) | 516 (9.7) | 197 (3.9) |
| Smoking status, n (%) | | | |
| Current smoker | 595 (5.3) | 347 (6.1) | 248 (4.4) |
| Non-smoker | 4,987 (44.0) | 2,103 (37.0) | 2,884 (51.0) |
| Former smoker | 5,761 (50.8) | 3,237 (56.9) | 2,524 (44.6) |
| Alcohol consumption, n (%) | | | |
| Almost every day | 2,474 (22.4) | 1,556 (27.8) | 918 (16.8) |
| 2–5 times a week | 2,878 (26.1) | 1,615 (28.9) | 1,263 (23.2) |
| 1–4 times a month | 2,912 (26.4) | 1,350 (24.2) | 1,562 (28.6) |
| Less than once a month | 1,405 (12.7) | 469 (8.4) | 936 (17.2) |
| Never | 1,371 (12.4) | 597 (10.7) | 774 (14.2) |
| Self-reported osteoporosis, n (%) | 1,603 (14.3) | 224 (4.0) | 1,379 (24.8) |
| Self-reported rheumatoid arthritis, n (%) | 449 (4.0) | 163 (2.9) | 286 (5.1) |
| Self-reported history of fractures since adulthood, n (%) | 2,052 (18.2) | 696 (12.3) | 1,356 (24.1) |
| Maternal fracture history, n (%) | 1,372 (12.5) | 611 (11.1) | 761 (13.8) |
| Corticosteroid use, n (%) | 1,601 (14.4) | 600 (10.8) | 1,001 (18.2) |
| Self-reported prior falls, n (%) | 601 (5.3) | 253 (4.5) | 348 (6.2) |
| Diabetes, n (%) | 2,403 (21.3) | 1,379 (24.3) | 1,024 (18.2) |
| The five-item Diener Satisfaction with Life Scale (SWLS, range 5–35), mean (SD) | 28.3 (5.9) | 28.8 (5.7) | 27.8 (6.0) |
| Center for Epidemiology Studies Depression 9 Scale (CES-D 9, range 0–27), mean (SD) | 4.6 (4.0) | 4.0 (3.6) | 5.2 (4.3) |
| Psychological distress (range 10–43), mean (SD) | 13.7 (4.2) | 13.3 (3.8) | 14.2 (4.5) |
| Nutritional risk (AB SCREEN II, range 0–48), mean (SD) | 39.0 (5.9) | 39.4 (5.7) | 38.6 (6.1) |
| Perceived mental health, n (%) | | | |
| Poor/ Fair | 441 (3.9) | 196 (3.5) | 245 (4.3) |
| Perceived health, n (%) | | | |
| Poor/ Fair | 940 (8.3) | 471 (8.3) | 469 (8.3) |
| Physical activity scale for the elderly score (PASE, range 0–629), mean (SD) | 116.1 (57.9) | 125.5 (60.7) | 106.7 (53.4) |
| Social isolation index (CLSA-SII, range 0–10), mean (SD) | 3.47 (1.40) | 3.26 (1.36) | 3.69 (1.40) |
| Grip strength (kg), mean (SD) | 31.92 (10.67) | 39.71 (8.49) | 23.74 (5.21) |
| Gait speed (m/s), mean (SD) | 0.93 (0.20) | 0.94 (0.20) | 0.91 (0.19) |
| DXA femoral neck BMD (g/cm$^2$), mean (SD) | 0.74 (0.13) | 0.79 (0.13) | 0.69 (0.11) |
| DXA total hip BMD (g/cm$^2$), mean (SD) | 0.90 (0.15) | 0.97 (0.14) | 0.82 (0.12) |

*(Continued)*

**Table 1.** (Continued)

| Baseline characteristic | All (n = 11,344) | Males (n = 5,688) | Females (n = 5,656) |
|---|---|---|---|
| Osteoporosis classification by DXA, n (%) | | | |
| Osteoporosis: DXA T-score ≤-2.5 | 804 (7.9) | 142 (2.8) | 662 (13.0) |
| Osteopenia: DXA T-score -1.0 to -2.5 | 4,442 (43.7) | 1,538 (30.3) | 2,904 (57.0) |
| Normal: DXA T-score ≥-1.0 | 4,929 (48.4) | 3,398 (66.9) | 1,531 (30.0) |
| Any self-reported prior fractures, n (%) | 271 (2.4) | 91 (1.6) | 180 (3.2) |
| **Three-year follow-up characteristic** | **All (n = 11,344)** | **Males (n = 5,688)** | **Females (n = 5,656)** |
| Grip strength (kg), mean (SD) | 30.89 (10.42) | 38.12 (8.68) | 23.09 (5.21) |
| Gait speed (m/s), mean (SD) | 0.89 (0.18) | 0.91 (0.18) | 0.87 (0.18) |
| DXA femoral neck BMD (g/cm$^2$), mean (SD) | 0.73 (0.13) | 0.78 (0.13) | 0.68 (0.11) |
| DXA total hip BMD (g/cm$^2$), mean (SD) | 0.89 (0.15) | 0.96 (0.14) | 0.81 (0.12) |
| Osteoporosis classification by DXA, n (%) | | | |
| Osteoporosis: DXA T-score ≤-2.5 | 841 (9.0) | 164 (3.4) | 677 (26.7) |
| Osteopenia: DXA T-score -1.0 to -2.5 | 4,210 (45.3) | 1,564 (32.8) | 2,646 (58.3) |
| Normal: DXA T-score ≥-1.0 | 4,248 (45.7) | 3,035 (63.7) | 1,213 (26.7) |
| Any self-reported incident fractures, n (%) | 345 (3.1) | 113 (2.0) | 232 (4.1) |

Abbreviations: SD = Standard Deviation; BMI = Body Mass Index; SWLS = Satisfaction with Life Scale; CES-D 9 = Center for Epidemiology Studies Depression 9 Scale;
AB SCREEN II = Abbreviated Seniors in the Community Risk Evaluation for Eating and Nutrition II; PASE = Physical Activity Scale for the Elderly score;
CLSA-SII = Canadian Longitudinal Study on Aging–Social Isolation Index; DXA = Dual-Energy X-ray absorptiometry; BMD = Bone Mineral Density
Non-weighted results.

**Table 2. Changes from baseline to post three-year follow-up.**

| Outcome variables | All (n = 11,344) | Males (n = 5,688) | Females (n = 5,656) |
|---|---|---|---|
| Grip strength, mean (SD) | | | |
| Absolute (kg) change | -1.33 (4.60) | -1.79 (5.24) | -0.83 (3.72) |
| Percentage change | -3.02 (16.65) | -3.78 (15.40) | -2.21 (17.87) |
| Gait speed, mean (SD) | | | |
| Absolute (m/s) change | -0.05 (0.17) | -0.05 (0.18) | -0.04 (0.17) |
| Percentage change | -3.06 (19.28) | -3.39 (19.09) | -3.13 (19.48) |
| Annualized absolute (g/cm$^2$) change in femoral neck BMD, mean (SD) | -0.004 (0.010) | -0.003 (0.010) | -0.004 (0.010) |
| Annualized percentage change in femoral neck BMD, mean (SD) | -0.47 (1.43) | -0.41 (1.33) | -0.52 (1.52) |
| Annualized absolute (g/cm$^2$) change in total hip BMD, mean (SD) | -0.005 (0.009) | -0.004 (0.009) | -0.006 (0.009) |
| Annualized percentage change in total hip BMD, mean (SD) | -0.57 (1.09) | -0.43 (1.02) | -0.71 (1.15) |
| Change for osteoporosis classification by DXA[a], n (%) | | | |
| Unchanged | 7,856 (88.3) | 4,126 (90.9) | 3,730 (85.6) |
| Worsened | 765 (8.6) | 306 (6.7) | 459 (10.5) |
| Improved | 277 (3.1) | 107 (2.4) | 170 (3.9) |
| Self-reported incident fractures[b], n (%) | | | |
| Yes | 345 (3.1) | 113 (2.0) | 232 (4.1) |
| No | 10,973 (96.9) | 5,563 (98.0) | 5,410 (95.9) |

Abbreviations: SD = Standard Deviation; BMD = Bone Mineral Density; SD = Standard Deviation; DXA = Dual-Energy X-ray absorptiometry
[a]Change for osteoporosis classification by DXA from baseline to follow-up were classified as follows: 1) unchanged (normal to normal, osteopenia to osteopenia, or osteoporosis to osteoporosis), 2) worsened (normal to osteopenia, normal to osteoporosis, or osteopenia to osteoporosis), and 3) improved (osteopenia to normal, osteoporosis to normal, or osteoporosis to osteopenia)
[b]Self-reported incident fractures from baseline to three-year follow-up were classified as follows: 1) yes (no prior fractures to incident fractures, or prior fractures to incident fractures) and 2) no (prior fractures to no incident fractures, or no prior fractures to no incident fractures)
Non-weighted results.

**Table 3. Results for multivariable linear regression models to determine the associations between baseline CLSA-SII and three-year changes in grip strength, gait speed, BMD, osteoporosis classification by DXA and self-reported incident fractures in all participants: Multiple imputation analysis (weighted case).**

| Three-year changes (dependent variables) | CLSA-SII (per unit one change) (independent variable) | |
|---|---|---|
| | Model 1 | Model 2 |
| | Unadjusted β or OR (95% CI) | Fully adjusted[a] β or OR (95% CI) |
| Grip strength (kg) | | |
| Absolute change | 0.064 (-0.025, 0.154) | 0.065 (-0.031, 0.162) |
| Percentage change | -0.050 (-0.381, 0.281) | 0.159 (-0.195, 0.513) |
| Gait speed (m/s) | | |
| Absolute change | 0.002 (-0.001, 0.005) | 0.001 (-0.003, 0.005) |
| Percentage change | 0.199 (-0.167, 0.565) | 0.117 (-0.310, 0.544) |
| Annualized absolute (g/cm$^2$) change in femoral neck BMD | -0.000 (-0.000, 0.000) | 0.000 (-0.000, 0.000) |
| Annualized percentage change in femoral neck BMD | -0.034 (-0.061, -0.007) | 0.004 (-0.025, 0.033) |
| Annualized absolute (g/cm$^2$) change in total hip BMD | -0.000 (-0.000, -0.000) | 0.000 (-0.000, 0.000) |
| Annualized percentage change in total hip BMD | -0.041 (-0.060, -0.020) | 0.007 (-0.015, 0.028) |
| Change for osteoporosis classification by DXA[b] | | |
| Unchanged | Ref | Ref |
| Worsened | 1.04 (0.97, 1.10) | 1.03 (0.94, 1.09) |
| Improved | 1.05 (0.96, 1.15) | 1.08 (0.97, 1.21) |
| Self-reported incident fractures[c] | | |
| Yes | 1.20 (1.08, 1.34) | 1.13 (1.01, 1.27) |
| No | Ref | Ref |

Abbreviations: CLSA-SII = Canadian Longitudinal Study on Aging–Social Isolation Index; SE = Standard Error; BMD = Bone Mineral Density; DXA = Dual-Energy X-ray absorptiometry; OR = Odds Ratio; 95% CI = 95% Confidence Interval

[a]Fully adjusted for age, sex, education, body mass index (BMI), total household income, smoking status, alcohol consumption, self-reported osteoporosis, self-reported rheumatoid arthritis, self-reported history of fractures since adulthood, maternal fracture history, corticosteroid use, self-reported prior falls, diabetes, DXA femoral neck BMD T-score, grip strength, gait speed, the five-item diener satisfaction with life scale (SWLS), centre for epidemiological studies depression scale (CES-D 9), psychological distress, nutritional risk (AB SCREEN II), perceived mental health, perceived health, and physical activity scale for the elderly (PASE)

Model 1 was unadjusted and Model 2 was fully adjusted.

[b]Change for osteoporosis classification by DXA from baseline to follow-up were classified as follows: 1) unchanged (normal to normal, osteopenia to osteopenia, or osteoporosis to osteoporosis), 2) worsened (normal to osteopenia, normal to osteoporosis, or osteopenia to osteoporosis), and 3) improved (osteopenia to normal, osteoporosis to normal, or osteoporosis to osteopenia)

[c]Self-reported incident fractures from baseline to three-year follow-up were classified as follows: 1) yes (no prior fractures to incident fractures, or prior fractures to incident fractures) and 2) no (prior fractures to no incident fractures, or no prior fractures to no incident fractures)

Weighted results, v1.2.

osteoporosis classification via DXA were not statistically significant in the univariate (Model 1) and multivariable (Model 2) analyses (Table 3). The results of the subgroup analysis were only significant for the association between CLSA-SII and femoral neck and total hip BMD in the unadjusted model (S2 Table).

**Table 4. Results of multivariable linear and logistic regression models of CLSA-SII in the three-year changes of absolute change in gait speed, annualized absolute (g/cm$^2$) in femoral neck BMD and self-reported incident fractures, when significant interaction terms were included in all participants.**

| | CLSA-SII (per unit one change) |
|---|---|
| **Three-year change** | **Model 3[a] CLSA-SII*PASE Δ interaction, β (95% CI)** |
| Absolute change in gait speed (m/s) | -0.000 (-0.000, -0.000)* |
| **Three-year change** | **Model 3[a] CLSA-SII*sex Δ interaction (Ref: Males), β (95% CI)** |
| Annualized absolute (g/cm$^2$) change in femoral neck BMD | 0.000 (0.000, 0.001)* |
| **Three-year change** | **Model 3[a] CLSA-SII*CES-D 9 Δ interaction, OR (95% CI)** |
| Self-reported incident fractures[b] | |
| Yes | 1.02 (1.00, 1.04)* |
| No | Ref |

Abbreviations: CLSA-SII = Canadian Longitudinal Study on Aging–Social Isolation Index; SE = Standard Error; BMD = Bone Mineral Density; PASE = Physical Activity Scale for Elderly; 95% CI = 95% Confidence Interval; OR = Odds Ratio; CES-D 9 = Centre for Epidemiology Study Depression Scale 9

[a]Model 3 was fully adjusted adding interaction terms to model 2, if significant interaction terms were found Adjustment for age, sex, education, body mass index (BMI), total household income, smoking status, alcohol consumption, self-reported osteoporosis, self-reported rheumatoid arthritis, self-reported history of fractures since adulthood, maternal fracture history, corticosteroid use, self-reported prior falls, diabetes, DXA femoral neck BMD T-score, grip strength, gait speed, the five-item diener satisfaction with life scale (SWLS), centre for epidemiology study depression scale 9 (CES-D 9), psychological distress, nutritional risk (AB SCREEN II), perceived mental health, perceived health, and physical activity scale for the elderly (PASE)

[b]Self-reported incident fractures from baseline to three-year follow-up were classified as follows: 1) yes (no prior fractures to incident fractures, or prior fractures to incident fractures) and 2) no (prior fractures to no incident fractures, or no prior fractures to no incident fractures)

*p<0.05.

Weighted results, v1.2.

Three significant interaction effects were found in the fully adjusted models (Model 3) (Table 4): 1) CLSA-SII and PASE for absolute change in gait speed (interaction β -0.000, 95% CI -0.000 to -0.000): the higher the PASE, the lower the CLSA-SII which indicates the significant interaction that less physically active individuals had greater social isolation ("Fig 1A") (i.e., CLSA-SII had a larger impact on absolute change in gait speed depending on one's physical activity level.); 2) CLSA-SII and sex for annualized absolute change in femoral neck BMD (ref males; interaction β 0.000, 95% CI 0.000 to 0.001): a decrease in BMD as CLSA-SII increased in males, but the opposite was true for females ("Fig 1B") (i.e., CLSA-SII had a larger impact on absolute change in femoral neck BMD depending on one's sex.); and 3) CLSA-SII and CES-D 9 for self-reported incident fractures (interaction OR 1.02, 95% CI 1.00–1.04): with greater CES-D 9, the probability of having self-reported incident fractures with greater CLSA-SII ("Fig 1C") (i.e., CLSA-SII had a larger impact on incident fractures depending on one's depressive symptom level.). All two-way interaction terms are shown in S3 Table. No collinearity between covariates were shown.

## Sensitivity analysis

All planned sensitivity analyses were similar to our main findings except we found a significant interaction between CLSA-SII and sex associated with percentage change in femoral neck BMD and annualized absolute change in total hip BMD (S4 and S5 Tables).

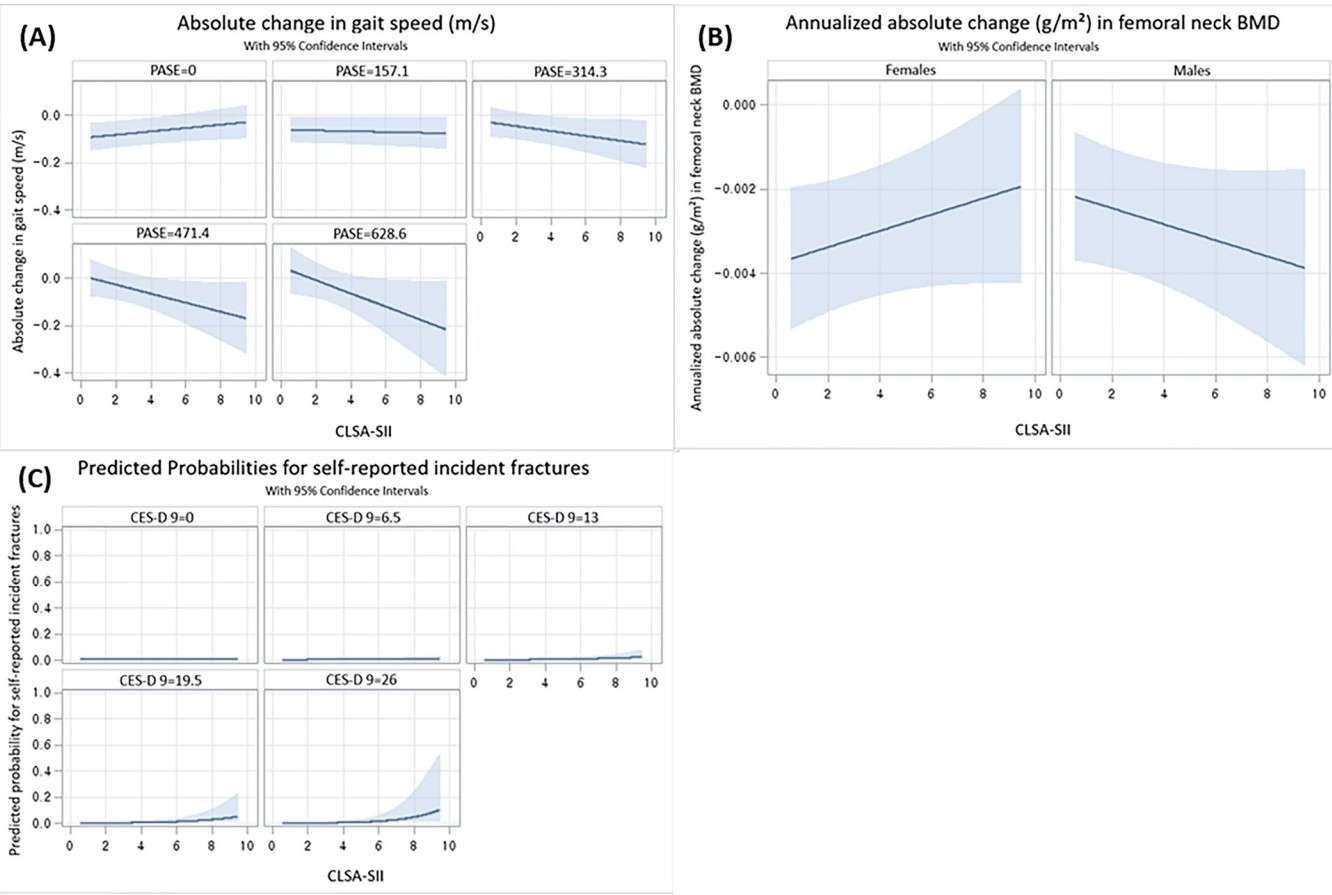

**Fig 1. How the predicted values for regression models of absolute change in gait speed, annualized absolute change (g/cm²) in femoral neck BMD and self-reported incident fractures depend on the model effects of CLSA-SII*PASE, CLSA-SII*sex or CLSA-SII*CES-D 9.** Abbreviations: BMD = Bone Mineral Density; CLSA-SII = Canadian Longitudinal Study on Aging–Social Isolation Index; PASE = Physical Activity Scale for the Elderly score; CES-D 9 = Centre for Epidemiological Studies Depression scale. (A) The absolute change in gait speed (m/s) as a function of CLSA-SII, which presents how the predicted response changes with increasing PASE. Depending on the PASE (range 0–629, higher score indicates greater level of physical activity), the horizontal axis represents CLSA-SII (range 0–10, higher score indicates greater social isolation and loneliness), and vertical axis represents the absolute change in gait speed. (B) The annualized absolute (g/cm²) change in femoral neck BMD as a function of CLSA-SII, which presents how the predicted response changes by sex. By sex (females and males), the horizontal axis represents CLSA-SII (range 0–10, high score indicates greater social isolation and loneliness), and the vertical axis represents the annualized absolute (g/cm²) change in femoral neck BMD. (C) The predicted probability of having self-reported incident fractures as a function of CLSA-SII, which presents how the predicted response changes with increasing CES-D 9. Depending on the CES-D 9 (range 0–27, higher score indicates more depression symptoms), the horizontal axis represents CLSA-SII (range 0–10, higher score indicates greater social isolation and loneliness), and the vertical axis represents the probability of self-reported incident fractures. Weighted results, v1.2.

## Discussion

Older adults with higher CLSA-SII had higher rates of incident fractures. This was particularly the case for older adults who have concurrent depression. Few studies have explored the association between social isolation and loneliness and fractures. The US study of longitudinal survey data of more than 22,000 individuals aged 65 years or older, reported that depression played an important role in the relationship between social isolation and falls [40], which were similar to our findings. Their results emphasized that social isolation was associated with depression, and subsequently increased the risk of falls, while our findings showed that greater depression scores could increase the risk of fractures in addition to high social isolation and loneliness. Indeed, previous studies have reported that social isolation or loneliness is correlated with increased severity of depression for older adults [41–43]. The English Longitudinal

Study of Ageing (ELSA) reported an increase in depression by 0.16 per 1 increase in loneliness (95% CI 0.13–0.19) [41]. Another UK cohort study reported that socially isolated older adults had 3.01 (95% CI 1.27–7.11) fold greater odds for depression [15]. Older adults with depression may not be able to properly manage or assess their health conditions because they are socially isolated due to limited social relationships [40], limiting their awareness of the risk of fractures. Additionally, loneliness may affect people biologically including stress responses (e.g., lower cortisol response to stress [44] and lower cortisol output [45]) which may increase the risk of depression, such as immune dysfunction or hypothalamus-cerebral-adrenal activity [41]. Among the symptoms of depression, hypogonadism, decreased growth hormone secretion or actions, and increased inflammatory cytokines such as interleukins 1 and 6 may be associated with low BMD [46, 47]. Antidepressant medications including selective serotonin reuptake inhibitors (SSRI) and tricyclics can result in fractures as well as falls [48]. We also acknowledge that most fractures may be caused by falls. Therefore, we adjusted for several covariates related to osteoporosis and fracture risk including falls (i.e., self-reported incident falls).

We measured femoral neck BMD and total hip BMD as BMD changes over time. However, our study did not observe a significant main effect between CLSA-SII and declines in BMD during the three-year follow-up, which corresponds with a cohort of community-dwelling older adults in the UK which reported no association between social isolation and BMD [15].

However, we found a significant interaction term between CLSA-SII and sex for annualized absolute change in femoral neck BMD, which indicates that sex may also play a significant role. In addition, when testing subgroup analysis by sex, as CLSA-SII increased, absolute change in femoral neck BMD increased in females and decreased in males. This may be because females are more likely to take an osteoporosis medication, which could affect their BMD. Males are often undertreated for osteoporosis and are less likely to be taking medication [49]. Our study did not adjust for osteoporosis medication use because we could not access data on these medications. Further research is needed on whether social isolation and loneliness can increase or decrease change in BMD by sex by considering the effects of osteoporosis medications. A significant interaction term between CLSA-SII and sex in femoral neck BMD was shown, however, one in total hip BMD was not shown. Total hip BMD is larger and contains both more cortical and triangular bones than femoral neck BMD, therefore, our results need to be interpreted carefully, given that total hip BMD is recommended over femoral neck BMD for assessing change in BMD according to the ISCD [50].

Assuming that weight-bearing physical activity and poor diet quality would affect BMD changes [15], our study included physical activity and nutritional risks as covariates, but there were no main effect statistically significant differences after full adjustments. Based on the hypothesis that social isolation and loneliness hinder physical activity and these effects may lead to BMD changes, the interaction term between CLSA-SII and physical activity was tested, but interactions were not significant. Our study also did not find significant associations between CLSA-SII and changes in grip strength and gait speed. However, the interaction term between CLSA-SII and PASE was significant for absolute change in gait speed. Participants with higher CLSA-SII were associated with negative (slower) impact on absolute change in gait speed when their physical activity levels were higher. The CLSA-Social Isolation Index (CLSA-SII) had a positive (higher) impact on gait speed when their physical activity levels were lower (i.e., when physical activity is low, social isolation and loneliness will exert a positive effect on gait speed. however, when physical activity is high, social isolation and loneliness will exert a negative impact on gait speed.). The results of the another ELSA study [51], showed that social isolation and loneliness were related to decreased gait speed during 6-year follow-up. The study reported that a lower socioeconomic status may be associated with physical

functional impairment and greater disability, which may also be associated with social relationships [51]. However, the participants in our sample relatively had high household income and education level. Although in univariate analyses, total household income and education level were associated with the CLSA-SII (all p-value < .001), in the multivariable analyses, these factors were not statistically significant. In contrast and like our results, a European longitudinal population-based cohort study of more than 20,000 males and females showed that loneliness was not associated with the 2-year incident sarcopenia [19]. Our study did not define sarcopenia by assessing grip strength and/or gait speed because categorizing change based on cut-offs suggested from various protocols [e.g., European Working Group on Sarcopenia in Old People (EWGSOP) [9, 10] and the Foundation for the National of Health (FNIH)] [52] meant that the people more likely to be reclassified were those who had a baseline grip strength and/or gait speed closet to the cut-off rather than those with the most change.

The previous studies have yielded conflicting results. As social isolation and loneliness are multifaceted and multidimensional concepts [53], they can be assessed with various tools rather than through a single construct [22, 24, 53]. For the same reason, our study also assessed social isolation and loneliness through multifaceted definitions of social isolation and loneliness [22, 24]. In addition, several previous studies have reported that depression [22, 53], life satisfaction [22, 53], physical activity [15, 39], nutritional risk [15], stress [54], or sleep [54] may act as moderators or mediators in relation to social isolation, and the results may vary depending on whether these effects are adjusted in relation to social isolation and loneliness. Moreover, since socioeconomic status (SES) factors such as economic or educational level may be related to social isolation and loneliness and deterioration of physical function [51], the results may vary depending on the characteristics of the study participants according to SES.

Strengths of this study include how our study comprehensively examined the association between social isolation index and grip strength and gait speed which can be markers for sarcopenia, BMD, and self-reported incident fractures in community-dwelling older adults. We evaluated social isolation and loneliness via three aspects (i.e., structural-objective, functional-objective and functional-subjective) supporting the multifaceted definition that encompasses the quantity and quality of social contact and interaction with others [22]. In addition, CLSA is a population-based national study, which minimizes selection bias that may result from the use of limited sampling frames by randomly recruiting a wide range of participants residing across Canada. Our study enabled our findings to better represent the population by using sample weights, a standard practice in the CLSA survey. Furthermore, this study assessed the interactions between social isolation index and related socio-psychological factors (i.e., depression and life satisfaction), behavioral factor (i.e., physical activity), and age and sex which may work as mechanisms in the relationship between social isolation/loneliness and health consequences. This enables us to confirm not only the associations between social isolation index and changes in grip strength, gait speed, BMD and fractures but also if age, sex, social-psychological factors and behavioral factor acted as moderators in the associations.

We acknowledge that this study has several limitations. As this study had a relatively short follow-up (i.e., three-years), no significant differences were identified in grip strength, gait speed and BMD. In addition, we acknowledge a "healthy" responder bias among participants in the CLSA because only people who had verbal or written communication skills in English or French or who had no issues in hearing or memory were eligible to participate. Furthermore, as fractures and factors for CLSA-SII were measured as self-reported, recall bias may decrease reliability. Since fractures were measured as self-reported and not adjudicated, participants who were not diagnosed with a fracture may be included, or those who were diagnosed with may not be included. In addition, fractures that occurred between 2015 and 2017 were

not measured, as fractures were only measured for 12 months (i.e., 2014–2015) from baseline and for 12 months (i.e., 2017–2018) from the three-year follow-up. As there were many missing values in-site specific fractures in our follow-up dataset, the fractures included in the study analysis could be not classified as major osteoporotic fractures (e.g., hip, humerus, spine, wrist, and pelvis). The CLSA did not acquire lumbar spine and wrist BMD, thus we could not use it. Lumbar spine BMD is more metabolically active, while the femoral neck BMD has more cortical bone and as a result is less metabolically active. Therefore, femoral neck BMD is less likely to show change [55]. Future studies using lumbar spine BMD may be more likely to demonstrate an association between social isolation/loneliness and BMD. Since most of the domains constructed in CLSA-SII used in our study were about social isolation, it is more likely to be interpreted in terms of social isolation rather than loneliness. Our study measured depression almost simultaneously with factors of CLSA-SII, so the directionality of the relationship cannot be confirmed. Finally, our study did not adjust for the use of medications such as osteoporosis treatment antidepressants or any medications related to falls. Thus, it should be noted that the association between observed incident fractures may be overestimated to a certain extent. Future studies should investigate the association between social isolation/loneliness and self-reported incident fractures by considering the effect of medications.

## Conclusions

Socially isolated and lonely older adults may be at risk of self-reported incident fractures in the presence of greater depression. The CLSA-Social Isolation Index (CLSA-SII) was not associated with the three-year changes in grip strength, gait speed and BMD for older adults. However, significant interactions between social isolation and physical activity in gait speed and between social isolation and sex in the absolute change of femoral neck BMD were shown. Future longitudinal studies with a longer observational period are needed to further investigate the associations.

## Supporting information

**S1 Checklist. STROBE statement—checklist of items that should be included in reports of *cohort studies.***
(DOCX)

**S1 Table. Univariate analyses of the CLSA-SII in all participants (n = 11,344).**
(DOCX)

**S2 Table. Subgroup analysis results by sex for multivariable linear regression models of CLSA-SII as risk factors of the three-year changes in grip strength, gait speed, BMD, osteoporosis classification by DXA, and self-reported incident fractures: Multiple imputation analysis (weighted case).**
(DOCX)

**S3 Table. Two-way interaction terms between CLSA-SII and age, sex, CES-D 9, SWLS and PASE in the three-year changes of grip strength, gait speed, BMD, osteoporosis classification by DXA and self-reported incident fractures in all participants.**
(DOCX)

**S4 Table. Results of multivariable linear, logistic, multinomial logistic regression models of CLSA-SII as risk factors of the three-year changes in grip strength, gait speed, BMD, osteoporosis by DXA and self-reported incident fractures in all participants.**
(DOCX)

**S5 Table. Two-way interaction terms between CLSA-SII and age, sex, CES-D, SWLS and PASE in the three-year changes of grip strength, gait speed, BMD, osteoporosis classification by DXA and self-reported incident fractures.**
(DOCX)

## Author Contributions

**Conceptualization:** Ahreum Lee, Caitlin McArthur, George Ioannidis, Alexandra Mayhew, Jonathan D. Adachi, Lauren E. Griffith, Lehana Thabane, Alexandra Papaioannou.

**Data curation:** Ahreum Lee.

**Formal analysis:** Ahreum Lee.

**Investigation:** Ahreum Lee, Caitlin McArthur, George Ioannidis, Alexandra Papaioannou.

**Methodology:** Ahreum Lee, Caitlin McArthur, George Ioannidis, Alexandra Mayhew, Alexandra Papaioannou.

**Project administration:** Ahreum Lee.

**Resources:** Ahreum Lee.

**Supervision:** Alexandra Papaioannou.

**Validation:** Ahreum Lee, Caitlin McArthur, George Ioannidis, Alexandra Mayhew, Jonathan D. Adachi, Lauren E. Griffith, Lehana Thabane, Alexandra Papaioannou.

**Visualization:** Ahreum Lee, Caitlin McArthur, George Ioannidis.

**Writing – original draft:** Ahreum Lee.

**Writing – review & editing:** Ahreum Lee, Caitlin McArthur, George Ioannidis, Alexandra Mayhew, Jonathan D. Adachi, Lauren E. Griffith, Lehana Thabane, Alexandra Papaioannou.

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
