## [Decision Letter · Decision Letter 0]

9 Aug 2023

PONE-D-23-20334Associations between social isolation and loneliness and changes in grip strength, gait speed, bone mineral density (BMD), and self-reported incident fractures among older adults: results from the Canadian Longitudinal Study on Aging (CLSA)PLOS ONE

Dear Dr. Lee,

Thank you for submitting your manuscript to PLOS ONE. After careful consideration, we feel that it has merit but does not fully meet PLOS ONE’s publication criteria as it currently stands. Therefore, we invite you to submit a revised version of the manuscript that addresses the points raised during the review process.

We look forward to receiving your revised manuscript.

Kind regards,

Mario Ulises Pérez-Zepeda, M.D., Ph.D.

Academic Editor

PLOS ONE

Journal Requirements:

Reviewers' comments:

Reviewer's Responses to Questions

**Comments to the Author**

1. Is the manuscript technically sound, and do the data support the conclusions?

Reviewer #1: Partly

Reviewer #2: Yes

2. Has the statistical analysis been performed appropriately and rigorously? 

Reviewer #1: Yes

Reviewer #2: Yes

3. Have the authors made all data underlying the findings in their manuscript fully available?

Reviewer #1: Yes

Reviewer #2: Yes

4. Is the manuscript presented in an intelligible fashion and written in standard English?

Reviewer #1: Yes

Reviewer #2: Yes

5. Review Comments to the Author

Reviewer #1: This paper reports the associations between social isolation and loneliness and changes in grip strength, gait speed, bone mineral density (BMD), and self-reported incident fractures among older adults: results from the Canadian Longitudinal Study on Aging (CLSA). This is an important clinic and research topic; I understood that the researchers are interested in to know if social isolation and loneliness contributed to fractures at follow-up in older adults. However, the use of several categorical and continues variables as exposition and response variables, resulted very confuse to me and the use of regression analysis deserve attention and improvements.

Due the complexity of the objective of the study, please clearly define the exposition variables and the response variables in methodology. Regard to the exposition variables, in the title and the objective, it seems that social isolation and loneliness are categorical variables; however, in methods and results, it appears as a continuous variable, mean the same? please clarify.

My major concern is related to the statistical analysis, due to the nature of several exposition variables and response variables (continues/categorical), should be interesting that authors state clearly which hypotheses variables were tested with lineal and logistic and multiple (lineal or logistic) regression analysis, and their corresponding covariates or adjusted variables, and all these should be reflected in the results section. Additionally, in the discussion section authors discussed the results of the univariates analysis (page 26; line). From my point of view, authors should report or should mention all related to the results of the simple (lineal o logistic) regression analysis and the criteria established to test the potential association between independent and dependent variables in the section of statistical analysis and results section. The selection of predictor variables for the final model do the author used Forward or Backward Stepwise, please clarify. Looking the covariates section, there is not any criteria that authors used to define the independent variables as covariates. Additionally, I just wondering if authors considered hand grip strength and gait speed as covariates (due collinearity) to test certain hypothesis variables. I notice that volunteers underwent to DXA analysis, I just wondering if total lean tissue or appendicular skeletal muscle were used as covariates.

What is the reason to use both absolute change and percentage change in the statistical analysis?

Reviewer #2: Dear authors. I congratulate you on the quality of your work. It is a well-written manuscript whose methodologies achieve the main objective. The paper has sufficient quality but also provides knowledge of interest for daily practice. For this reason, I recommend that your work be published in PLos.

6. PLOS authors have the option to publish the peer review history of their article (what does this mean?). If published, this will include your full peer review and any attached files.

Reviewer #1: No

Reviewer #2: No

---

## [Author Response · Author response to Decision Letter 0]

20 Sep 2023

PONE-D-23-20334

Associations between social isolation and loneliness and changes in grip strength, gait speed, bone mineral density (BMD), and self-reported incident fractures among older adults: results from the Canadian Longitudinal Study on Aging (CLSA)

PLOS ONE

Reviewers' comments:

Reviewer's Responses to Questions

Comments to the Author

1. Is the manuscript technically sound, and do the data support the conclusions?

Reviewer #1: Partly

Reviewer #2: Yes

2. Has the statistical analysis been performed appropriately and rigorously?

Reviewer #1: Yes

Reviewer #2: Yes

3. Have the authors made all data underlying the findings in their manuscript fully available?

Reviewer #1: Yes

Reviewer #2: Yes

4. Is the manuscript presented in an intelligible fashion and written in standard English?

Reviewer #1: Yes

Reviewer #2: Yes

5. Review Comments to the Author

Reviewer #1: This paper reports the associations between social isolation and loneliness and changes in grip strength, gait speed, bone mineral density (BMD), and self-reported incident fractures among older adults: results from the Canadian Longitudinal Study on Aging (CLSA). This is an important clinic and research topic; I understood that the researchers are interested in to know if social isolation and loneliness contributed to fractures at follow-up in older adults. However, the use of several categorical and continues variables as exposition and response variables, resulted very confuse to me and the use of regression analysis deserve attention and improvements.

We agree with the reviewer’s comments. We have added the following statement to the methods section:

“Based on the findings of previous epidemiological studies, we included the following baseline covariates related to the associations between baseline social isolation and loneliness and changes in grip strength, gait speed, BMD, and self-reported incident fractures at year 3: age group (65-74 years, ≥75 years), sex (males, females), education (less than secondary, secondary, post-secondary), body mass index (BMI, kg/m2), total household income (less than $20,000, $20,000-$49,999, $50,000-$99,999, $100,000-$149,999, ≥$150,000), smoking status (current smoker, non-smoker, former smoker), alcohol consumption (almost every day, 2-5 times a week, 1-4 times a month, less than once a month, never), self-reported osteoporosis (yes, no), self-reported rheumatoid arthritis (yes, no), self-reported history of fractures since adulthood (yes, no), maternal fractures history (yes, no), corticosteroid use (yes, no), self-reported prior falls (yes, no), diabetes (yes, no), DXA femoral neck BMD T-score (SD), grip strength (kg), gait speed (m/s), the five-item satisfaction with life scale [(SWLS), range 5-35, where higher scores indicate greater life satisfaction],(31, 32) centre for epidemiology studies depression 9 Scale [(CES-D 9), range 0-27 where higher scores indicate greater depression],(33) psychological distress (range 10-43, where higher scores indicate higher mental distress),(34, 35) nutritional risk [(AB SCREEN II)range 0-48 where higher scores indicate a higher level of nutritional risk],(36) perceived mental health, perceived health, and physical activity scale for the elderly score (PASE) (range 0-629, where higher scores indicate greater physical activity.(37) Given that a single-item loneliness question that is a part of the CES-D 10 was included in the CLSA-SII instrument, a total of nine items (e.g., easily bothered and feel fearful or tearful) except for loneliness was included in CES-D 9.(22) This was based on the evidence that SWLS, perceived health, perceived mental health and CES-D 9 correlated with the CLSA-SII from the previous study by Wister et al.,(22). In addition, we included other variables related to osteoporosis or fractures, such as self-reported osteoporosis and self-reported history of fractures since adulthood, as covariates.” (Page 9, Line 193)

“We hypothesized that socially isolated and lonely older adults may have greater reductions in grip strength, gait speed, or (femoral neck and total hip) BMD or had an increased risk of fractures.” (Page 10, Line 225)

“To test these hypotheses, we conducted each of the following multivariable analyses: 1) independent (exposure) variable: CLSA-SII (continuous), 2) outcome (response) variables: a) changes in grip strength (continuous, linear), b) changes in gait speed (continuous, linear), c) changes in BMD (continuous, linear) and change for osteoporosis classification by DXA (category, multinomial), d) self-reported incident fractures (binary, logistic). Model 1 was unadjusted and Model 2 was a fully adjusted multivariable linear or logistic and multinomial logistic regression model. In Model 2, all outcome variables were adjusted for all the same covariates.” (Page 11, Line 228)

Due the complexity of the objective of the study, please clearly define the exposition variables and the response variables in methodology. 

We agree with the reviewer’s comments. We have added the following statement to the methods section:

“Based on the findings of previous epidemiological studies, we included the following baseline covariates related to the associations between baseline social isolation and loneliness and changes in grip strength, gait speed, BMD, and self-reported incident fractures at year 3: age group (65-74 years, ≥75 years), sex (males, females), education (less than secondary, secondary, post-secondary), body mass index (BMI, kg/m2), total household income (less than $20,000, $20,000-$49,999, $50,000-$99,999, $100,000-$149,999, ≥$150,000), smoking status (current smoker, non-smoker, former smoker), alcohol consumption (almost every day, 2-5 times a week, 1-4 times a month, less than once a month, never), self-reported osteoporosis (yes, no), self-reported rheumatoid arthritis (yes, no), self-reported history of fractures since adulthood (yes, no), maternal fractures history (yes, no), corticosteroid use (yes, no), self-reported prior falls (yes, no), diabetes (yes, no), DXA femoral neck BMD T-score (SD), grip strength (kg), gait speed (m/s), the five-item satisfaction with life scale [(SWLS), range 5-35, where higher scores indicate greater life satisfaction],(31, 32) centre for epidemiology studies depression 9 Scale [(CES-D 9), range 0-27 where higher scores indicate greater depression],(33) psychological distress (range 10-43, where higher scores indicate higher mental distress),(34, 35) nutritional risk [(AB SCREEN II)range 0-48 where higher scores indicate a higher level of nutritional risk],(36) perceived mental health, perceived health, and physical activity scale for the elderly score (PASE) (range 0-629, where higher scores indicate greater physical activity.(37) Given that a single-item loneliness question that is a part of the CES-D 10 was included in the CLSA-SII instrument, a total of nine items (e.g., easily bothered and feel fearful or tearful) except for loneliness was included in CES-D 9.(22) This was based on the evidence that SWLS, perceived health, perceived mental health and CES-D 9 correlated with the CLSA-SII from the previous study by Wister et al.,(22). In addition, we included other variables related to osteoporosis or fractures, such as self-reported osteoporosis and self-reported history of fractures since adulthood, as covariates.” (Page 9, Line 193)

“We hypothesized that socially isolated and lonely older adults may have greater reductions in grip strength, gait speed, or (femoral neck and total hip) BMD or had an increased risk of fractures.” (Page 10, Line 225) 

“To test these hypotheses, we conducted each of the following multivariable analyses: 1) independent (exposure) variable: CLSA-SII (continuous), 2) outcome (response) variables: a) changes in grip strength (continuous, linear), b) changes in gait speed (continuous, linear), c) changes in BMD (continuous, linear) and change for osteoporosis classification by DXA (category, multinomial), d) self-reported incident fractures (binary, logistic). Model 1 was unadjusted and Model 2 was a fully adjusted multivariable linear or logistic and multinomial logistic regression model. In Model 2, all outcome variables were adjusted for all the same covariates.” (Page 11, Line 228)

Regard to the exposition variables, in the title and the objective, it seems that social isolation and loneliness are categorical variables; however, in methods and results, it appears as a continuous variable, mean the same? Please clarify.

We agree with the reviewer’s comments. We assessed social isolation and loneliness as a continuous variable using the social isolation index. We replace the expression of social isolation and loneliness with the social isolation index (The results analyzed or assessment of social isolation and loneliness in our study are revised to social isolation index). 

We have revised the following statement to the title:

“Associations between Social Isolation Index and changes in grip strength, gait speed, bone mineral density (BMD), and self-reported incident fractures among older adults: results from the Canadian Longitudinal Study on Aging (CLSA)” (Page 1, Line 1)

We have revised the following statement to the introduction section:

“Therefore, our aim was to explore whether baseline social isolation index was associated with changes in grip strength, gait speed, BMD, and self-reported incident fractures among community-living older adults in Canada at the three-year follow-up.” (Page 4, Line 89) 

We have revised the following statement to the discussion section:

“Older adults with higher CLSA-SII had higher rates of incident fractures.” (Page 24, Line 404)

“Participants with higher CLSA-SII were associated with negative (slower) impact on absolute change in gait speed when their physical activity levels were higher.” (Page 26, Line 456)

“The CLSA-Social Isolation Index (CLSA-SII) had a positive (higher) impact on gait speed when their physical activity levels were lower (i.e., when physical activity is low, social isolation and loneliness will exert a positive effect on gait speed.” (Page 26, Line 457)

“Strengths of this study include how our study comprehensively examined the association between social isolation index and grip strength and gait speed which can be markers for sarcopenia, BMD, and self-reported incident fractures in community-dwelling older adults.” (Page 28, Line 489)

“Furthermore, this study assessed the interactions between social isolation index and related socio-psychological factors (i.e., depression and life satisfaction), behavioral factor (i.e., physical activity), and age and sex which may work as mechanisms in the relationship between social isolation/loneliness and health consequences. This enables us to confirm not only the associations between social isolation index and changes in grip strength, gait speed, BMD and fractures but also if age, sex, social-psychological factors and behavioral factor acted as moderators in the associations.” (Page 28, Line 498)

We have revised the following statement to the conclusion section:

“The CLSA-Social Isolation Index (CLSA-SII) was not associated with the three-year changes in grip strength, gait speed and BMD for older adults.” (Page 30, Line 534)

My major concern is related to the statistical analysis, due to the nature of several exposition variables and response variables (continues/categorical), should be interesting that authors state clearly which hypotheses variables were tested with lineal and logistic and multiple (lineal or logistic) regression analysis, and their corresponding covariates or adjusted variables, and all these should be reflected in the results section. 

We agree with the reviewer’s comments. We have added the following statement to the methods section:

“Based on the findings of previous epidemiological studies, we included the following baseline covariates related to the associations between baseline social isolation and loneliness and changes in grip strength, gait speed, BMD, and self-reported incident fractures at year 3: age group (65-74 years, ≥75 years), sex (males, females), education (less than secondary, secondary, post-secondary), body mass index (BMI, kg/m2), total household income (less than $20,000, $20,000-$49,999, $50,000-$99,999, $100,000-$149,999, ≥$150,000), smoking status (current smoker, non-smoker, former smoker), alcohol consumption (almost every day, 2-5 times a week, 1-4 times a month, less than once a month, never), self-reported osteoporosis (yes, no), self-reported rheumatoid arthritis (yes, no), self-reported history of fractures since adulthood (yes, no), maternal fractures history (yes, no), corticosteroid use (yes, no), self-reported prior falls (yes, no), diabetes (yes, no), DXA femoral neck BMD T-score (SD), grip strength (kg), gait speed (m/s), the five-item satisfaction with life scale [(SWLS), range 5-35, where higher scores indicate greater life satisfaction],(31, 32) centre for epidemiology studies depression 9 Scale [(CES-D 9), range 0-27 where higher scores indicate greater depression],(33) psychological distress (range 10-43, where higher scores indicate higher mental distress),(34, 35) nutritional risk [(AB SCREEN II)range 0-48 where higher scores indicate a higher level of nutritional risk],(36) perceived mental health, perceived health, and physical activity scale for the elderly score (PASE) (range 0-629, where higher scores indicate greater physical activity.(37) Given that a single-item loneliness question that is a part of the CES-D 10 was included in the CLSA-SII instrument, a total of nine items (e.g., easily bothered and feel fearful or tearful) except for loneliness was included in CES-D 9.(22) This was based on the evidence that SWLS, perceived health, perceived mental health and CES-D 9 correlated with the CLSA-SII from the previous study by Wister et al.,(22). In addition, we included other variables related to osteoporosis or fractures, such as self-reported osteoporosis and self-reported history of fractures since adulthood, as covariates.” (Page 9, Line 193)

“We hypothesized that socially isolated and lonely older adults may have greater reductions in grip strength, gait speed, or (femoral neck and total hip) BMD or had an increased risk of fractures.” (Page 10, Line 225) 

“To test these hypotheses, we conducted each of the following multivariable analyses: 1) independent (exposure) variable: CLSA-SII (continuous), 2) outcome (response) variables: a) changes in grip strength (continuous, linear), b) changes in gait speed (continuous, linear), c) changes in BMD (continuous, linear) and change for osteoporosis classification by DXA (category, multinomial), d) self-reported incident fractures (binary, logistic). Model 1 was unadjusted and Model 2 was a fully adjusted multivariable linear or logistic and multinomial logistic regression model. In Model 2, all outcome variables were adjusted for all the same covariates.” (Page 11, Line 228)

Additionally, in the discussion section authors discussed the results of the univariates analysis (page 26; line). From my point of view, authors should report or should mention all related to the results of the simple (lineal o logistic) regression analysis and the criteria established to test the potential association between independent and dependent variables in the section of statistical analysis and results section. 

We agree with the reviewer’s comments. We have added the following statement to the methods section:

“We hypothesized that socially isolated and lonely older adults may have greater reductions in grip strength, gait speed, or (femoral neck and total hip) BMD or had an increased risk of fractures." (Page 10, Line 225)

“To test these hypotheses, we conducted each of the following multivariable analyses: 1) independent (exposure) variable: CLSA-SII (continuous), 2) outcome (response) variables: a) changes in grip strength (continuous, linear), b) changes in gait speed (continuous, linear), c) changes in BMD (continuous, linear) and change for osteoporosis classification by DXA (category, multinomial), d) self-reported incident fractures (binary, logistic). Model 1 was unadjusted and Model 2 was a fully adjusted multivariable linear or logistic and multinomial logistic regression model. In Model 2, all outcome variables were adjusted for all the same covariates.” (Page 11, Line 228)

We have added the following statement to the results section:

“Socially isolated and lonely older participants had a greater likelihood of self-reported incident fractures (unadjusted OR 1.20, 95% CI 1.08-1.34) (Model 1).” (Page 12, Line 275)

“The associations between CLSA-SII and absolute and percentage change in grip strength and gait speed, annualized absolute and percentage changes in femoral neck and total hip BMD and osteoporosis classification via DXA were not statistically significant in the univariate (Model 1) and multivariable (Model 2) analyses (Table 3).” (Page 13, Line 279)

We have revised the following statement to the discussion section:

“The study reported that a lower socioeconomic status may be associated with physical functional impairment and greater disability, which may also be associated with social relationships.(51) However, the participants in our sample relatively had high household income and education level. Although in univariate analyses, total household income and education level were associated with the CLSA-SII (all p-value<.001), in the multivariable analyses, these factors were not statistically significant.” (Page 27, Line 463)

The selection of predictor variables for the final model do the author used Forward or Backward Stepwise, please clarify. Looking the covariates section, there is not any criteria that authors used to define the independent variables as covariates. 

Thank you for your comment. 

Our study aimed to determine whether social isolation/loneliness is associated with grip strength, gait speed, and BMD changes, rather than identifying risk factors related to the associations between social isolation/loneliness and the outcomes. Therefore, in selecting covariates, we considered factors related to the associations based on previous epidemiological studies, rather than using a forward/backward stepwise approach. However, we have included Supplementary Table 1, which presents the results of univariate analyses of the CLSA-SII in all participants. 

We have added the following statement to the methods section:

“Based on the findings of previous epidemiological studies, we included the following baseline covariates related to the associations between baseline social isolation and loneliness and changes in grip strength, gait speed, BMD, and self-reported incident fractures at year 3: age group (65-74 years, ≥75 years), sex (males, females), education (less than secondary, secondary, post-secondary), body mass index (BMI, kg/m2), total household income (less than $20,000, $20,000-$49,999, $50,000-$99,999, $100,000-$149,999, ≥$150,000), smoking status (current smoker, non-smoker, former smoker), alcohol consumption (almost every day, 2-5 times a week, 1-4 times a month, less than once a month, never), self-reported osteoporosis (yes, no), self-reported rheumatoid arthritis (yes, no), self-reported history of fractures since adulthood (yes, no), maternal fractures history (yes, no), corticosteroid use (yes, no), self-reported prior falls (yes, no), diabetes (yes, no), DXA femoral neck BMD T-score (SD), grip strength (kg), gait speed (m/s), the five-item satisfaction with life scale [(SWLS), range 5-35, where higher scores indicate greater life satisfaction],(31, 32) centre for epidemiology studies depression 9 Scale [(CES-D 9), range 0-27 where higher scores indicate greater depression],(33) psychological distress (range 10-43, where higher scores indicate higher mental distress),(34, 35) nutritional risk [(AB SCREEN II)range 0-48 where higher scores indicate a higher level of nutritional risk),(36) perceived mental health, perceived health, and physical activity scale for the elderly score (PASE) (range 0-629, where higher scores indicate greater physical activity.(37) Given that a single-item loneliness question that is a part of the CES-D 10 was included in the CLSA-SII instrument, a total of nine items (e.g., easily bothered and feel fearful or tearful) except for loneliness was included in CES-D 9.(22) This was based on the evidence that SWLS, perceived health, perceived mental health and CES-D 9 correlated with the CLSA-SII from the previous study by Wister et al.,(22). In addition, we included other variables related to osteoporosis or fractures, such as self-reported osteoporosis and self-reported history of fractures since adulthood, as covariates.” (Page 9, Line 193)

“We used T-Test (binary, dichotomy), Welch’s ANOVA (category), and Pearson correlation (continuous) for univariate analyses of the CLSA-SII in all participants.” (Page 10, Line 227)

We have added the following statement to the results section:

“Univariate analyses of the CLSA-SII in all participants are shown in Supplementary Table S1. (Page 12, Line 265)

Additionally, I just wondering if authors considered hand grip strength and gait speed as covariates (due collinearity) to test certain hypothesis variables. 

Thank you for your comment. We included grip strength and gait speed at baseline as covariates in multivariable analyses. We omitted to describe including grip strength and gait speed at baseline as covariates in our manuscript. We also tested collinearity between grip strength and gait speed. However, we could not find statistical significance (tolerance=0.832, VIF=1.203). 

We added the following statement to the methods section:

“Based on the findings of previous epidemiological studies, we included the following baseline covariates related to the associations between baseline social isolation and loneliness and changes in grip strength, gait speed, BMD, and self-reported incident fractures at year 3: age group (65-74 years, ≥75 years), sex (males, females), education (less than secondary, secondary, post-secondary), body mass index (BMI, kg/m2), total household income (less than $20,000, $20,000-$49,999, $50,000-$99,999, $100,000-$149,999, ≥$150,000), smoking status (current smoker, non-smoker, former smoker), alcohol consumption (almost every day, 2-5 times a week, 1-4 times a month, less than once a month, never), self-reported osteoporosis (yes, no), self-reported rheumatoid arthritis (yes, no), self-reported history of fractures since adulthood (yes, no), maternal fractures history (yes, no), corticosteroid use (yes, no), self-reported prior falls (yes, no), diabetes (yes, no), DXA femoral neck BMD T-score (SD), grip strength (kg), gait speed (m/s), the five-item satisfaction with life scale [(SWLS), range 5-35, where higher scores indicate greater life satisfaction],(31, 32) centre for epidemiology studies depression 9 Scale [(CES-D 9), range 0-27 where higher scores indicate greater depression],(33) psychological distress (range 10-43, where higher scores indicate higher mental distress),(34, 35) nutritional risk [(AB SCREEN II)range 0-48 where higher scores indicate a higher level of nutritional risk),(36) perceived mental health, perceived health, and physical activity scale for the elderly score (PASE) (range 0-629, where higher scores indicate greater physical activity.(37) Given that a single-item loneliness question that is a part of the CES-D 10 was included in the CLSA-SII instrument, a total of nine items (e.g., easily bothered and feel fearful or tearful) except for loneliness was included in CES-D 9.(22) This was based on the evidence that SWLS, perceived health, perceived mental health and CES-D 9 correlated with the CLSA-SII from the previous study by Wister et al.,(22). In addition, we included other variables related to osteoporosis or fractures, such as self-reported osteoporosis and self-reported history of fractures since adulthood, as covariates.” (Page 9, Line 193)

“We also tested collinearity between covariates (e.g., grip strength and gait speed).” (Page 11, Line 243)

We have added the following statement to the results section:

No collinearity between covariates were shown. (Page 14, Line 298)

We have added the following statement to the table section:

“aFully adjusted for age, sex, education, body mass index (BMI), total household income, smoking status, alcohol consumption, self-reported osteoporosis, self-reported rheumatoid arthritis, self-reported history of fractures since adulthood, maternal fracture history, corticosteroid use, self-reported prior falls, diabetes, DXA femoral neck BMD T-score, grip strength, gait speed, the five-item diener satisfaction with life scale (SWLS), centre for epidemiological studies depression scale (CES-D 9), psychological distress, nutritional risk (AB SCREEN II), perceived mental health, perceived health, and physical activity scale for the elderly (PASE)” (Page 20, Line 328)

“Adjustment for age, sex, education, body mass index (BMI), total household income, smoking status, alcohol consumption, self-reported osteoporosis, self-reported rheumatoid arthritis, self-reported history of fractures since adulthood, maternal fracture history, corticosteroid use, self-reported prior falls, diabetes, DXA femoral neck BMD T-score, grip strength, gait speed, the five-item diener satisfaction with life scale (SWLS), centre for epidemiology study depression scale 9 (CES-D 9), psychological distress, nutritional risk (AB SCREEN II), perceived mental health, perceived health, and physical activity scale for the elderly (PASE)” (Page 21, Line 357)

I notice that volunteers underwent to DXA analysis, I just wondering if total lean tissue or appendicular skeletal muscle were used as covariates.

Thank you for your comment. We did not include total lean tissue or appendicular skeletal muscle as covariates in multivariable analyses. According to SDOC guidelines, lean mass measured by DXA was not associated with incident adverse health-related outcomes with or without adjustment for body size. Instead, we included BMI as a covariate in multivariable analyses. 

What is the reason to use both absolute change and percentage change in the statistical analysis?

Thank you for your comment. We have added the following statement to the methods section:

“For bone density, the smallest detectable difference in absolute measurements (g/cm2) remains consistent regardless of the baseline BMD, while the percentage change is influenced by both the baseline bone density and age.(27) We assessed BMD changes by considering both absolute and percentage changes in BMD according to the International Society for Clinical Densitometry (ISCD).(28) The absolute change represents the actual numeric variation in BMD over a specific period, while the percentage change is expressed as a ratio relative to the baseline BMD change rate, enabling the comparison of BMD changes among individuals with different baseline BMD values.(28) It is important to note that individuals with a very low baseline BMD may have a larger percentage change in BMD compared to those with a higher baseline BMD due to the same absolute BMD change.(29)” (Page 7, Line 151)

Reviewer #2: Dear authors. I congratulate you on the quality of your work. It is a well-written manuscript whose methodologies achieve the main objective. The paper has sufficient quality but also provides knowledge of interest for daily practice. For this reason, I recommend that your work be published in PLos.

6. PLOS authors have the option to publish the peer review history of their article (what does this mean?). If published, this will include your full peer review and any attached files.

Do you want your identity to be public for this peer review? For information about this choice, including consent withdrawal, please see our Privacy Policy.

Yes

---

## [Editor Report · Decision Letter 1]

28 Sep 2023

Associations between Social Isolation Index and changes in grip strength, gait speed, bone mineral density (BMD), and self-reported incident fractures among older adults: results from the Canadian Longitudinal Study on Aging (CLSA)

PONE-D-23-20334R1

Dear Dr. Lee,

We’re pleased to inform you that your manuscript has been judged scientifically suitable for publication and will be formally accepted for publication once it meets all outstanding technical requirements.

Kind regards,

Mario Ulises Pérez-Zepeda, M.D., Ph.D.

Academic Editor

PLOS ONE
---

## [Editor Report · Acceptance letter]

9 Oct 2023

PONE-D-23-20334R1 

Associations between Social Isolation Index and changes in grip strength, gait speed, bone mineral density (BMD), and self-reported incident fractures among older adults: results from the Canadian Longitudinal Study on Aging (CLSA) 

Dear Dr. Lee:

I'm pleased to inform you that your manuscript has been deemed suitable for publication in PLOS ONE. Congratulations! Your manuscript is now with our production department. 

Kind regards, 

on behalf of

Dr. Mario Ulises Pérez-Zepeda 

Academic Editor

PLOS ONE